# Hotspots and super-spreaders: Modelling fine-scale malaria parasite transmission using mosquito flight behaviour

Luigi Sedda[1]*, Robert S. McCann[2,3,4], Alinune N. Kabaghe[3], Steven Gowelo[2,3,5], Monicah M. Mburu[2,3], Tinashe A. Tizifa[3,6], Michael G. Chipeta[3,7], Henk van den Berg[2], Willem Takken[2], Michèle van Vugt[6], Kamija S. Phiri[3], Russell Cain[1], Julie-Anne A. Tangena[8], Christopher M. Jones[7,8]*

1 Lancaster Ecology and Epidemiology Group, Lancaster Medical School, Lancaster University, United Kingdom, 2 Laboratory of Entomology, Wageningen University & Research, Wageningen, The Netherlands, 3 School of Global and Public Health, Kamuzu University of Health Sciences, Blantyre, Malawi, 4 Center for Vaccine Development and Global Health, University of Maryland School of Medicine, Baltimore, Maryland, United States of America, 5 MAC Communicable Diseases Action Centre, Kamuzu University of Health Sciences, Blantyre, Malawi, 6 Center for Tropical Medicine and Travel Medicine, University of Amsterdam, The Netherlands, 7 Malawi-Liverpool-Wellcome Trust Clinical Research Programme, Blantyre, Malawi, 8 Vector Biology Department, Liverpool School of Tropical Medicine, Liverpool, United Kingdom

* l.sedda@lancaster.ac.uk (LS); chris.jones@lstmed.ac.uk (CMJ)

**Data Availability Statement:** All relevant data and software required to run the analysis are located in the following Lancaster University PURE repository: https://doi.org/10.17635/lancaster/

## Abstract

Malaria hotspots have been the focus of public health managers for several years due to the potential elimination gains that can be obtained from targeting them. The identification of hotspots must be accompanied by the description of the overall network of stable and unstable hotspots of malaria, especially in medium and low transmission settings where malaria elimination is targeted. Targeting hotspots with malaria control interventions has, so far, not produced expected benefits. In this work we have employed a mechanistic-stochastic algorithm to identify clusters of super-spreader houses and their related stable hotspots by accounting for mosquito flight capabilities and the spatial configuration of malaria infections at the house level. Our results show that the number of super-spreading houses and hotspots is dependent on the spatial configuration of the villages. In addition, super-spreaders are also associated to house characteristics such as livestock and family composition. We found that most of the transmission is associated with winds between 6pm and 10pm although later hours are also important. Mixed mosquito flight (downwind and upwind both with random components) were the most likely movements causing the spread of malaria in two out of the three study areas. Finally, our algorithm (named MALSWOTS) provided an estimate of the speed of malaria infection progression from house to house which was around 200–400 meters per day, a figure coherent with mark-release-recapture studies of *Anopheles* dispersion. Cross validation using an out-of-sample procedure showed accurate identification of hotspots. Our findings provide a significant contribution towards the identification and development of optimal tools for efficient and effective spatio-temporal targeted malaria interventions over potential hotspot areas.

researchdata/539 (data) https://doi.org/10.17635/lancaster/researchdata/511 (software).

**Funding:** This work was funded by the Wellcome Trust Seed Award to CMJ (212501/Z/18/Z). Data were collected as part of the Majete Malaria Project, which was funded by Stichting Dioraphte, The Netherlands, Grant Number 13050800, awarded to WT. LS is also supported by the Wellcome Trust NIHR–Wellcome Partnership for Global Health Research Collaborative Award, CEASE (220870/Z/20/Z), by EPSRC, DSNE, (EP/R01860X/1), and by the Academy of Medical Sciences GCRF Networking Grant Scheme (GCRFNGR7/1329). JAT is supported by the Medical Research Council Skills Development Fellowship. RM received additional support from National Institute of Health award no. K01TW011770. The funders had no role in study design, data collection and analysis, decision to publish, or preparation of the manuscript.

**Competing interests:** The authors have declared that no competing interests exist.

## Author summary

The dispersal of infectious *Anopheles* mosquitoes is critical to determining the geographical range over which malaria parasites are transmitted between human hosts and mosquito vectors. Malaria rates in the human population vary over space and time and are often characterised by hotspots, where disproportionately few hosts or individuals contribute to malaria transmission. Here, we present an approach to determine the location of malaria hotspots and super spreader houses based on modelling infectious mosquito movements from house to house using infection and wind data collected as part of a wider malaria study in southern Malawi. From our model, we show that it is possible to determine key components of malaria transmission including the identification of stable and unstable malaria hotspots (including super spreader houses), how quickly malaria spreads between households, quantify the importance of village configuration on malaria spread and identify the most important wind types in the local ecological setting. We conclude that it is possible to determine networks of mosquito-borne infection from combining infection and wind data. The identification of malaria hotspots presents an opportunity to target malaria control efforts in areas where malaria is disproportionately high.

## Introduction

Targeted malaria control is an important and potentially cost-effective approach [1,2]. Developing effective strategies for implementing malaria interventions on a targeted basis is particularly important in an era where countries are striving to achieve malaria elimination targets using limited funding and when progress in reducing malaria cases has stalled over the last 5 years [3]. One critical component of a targeted control strategy is the accurate delineation of entomological and epidemiological hotspots (areas where vector/s or disease rates exceed a local average value [4]), that in hypoendemic and mesoendemic malaria regions are often characterised by a combination of stable hotspots (persisting across time), unstable hotspots (seasonally occurring but not consistent geographically) and low transmission clusters (persistent across time but not reaching hotspot threshold levels) within different spatial scales. Therefore, defining and delineating hotspots according to available vector and malaria data is difficult and this is compounded by a lack of information on mosquito and people movement and the interaction between vectors and hosts [5], and consequently requires fast and flexible malaria surveillance systems with digital tools for optimising sampling locations, collection and analysis of key malaria data [3,5].

For these reasons, understanding and quantifying mosquito movement is of key importance for identifying hotspot networks, forecasting fine-scale malaria transmission and for geographically targeted interventions, especially in countries where there is focal and heterogeneous transmission [5–7]. Super-spreaders are characterised by temporal and spatial scales intrinsically associated to the stability and size of hotspots areas. Hence, super-spreaders need to be considered in any rational design of vector control programmes. There is not a conventional definition of super-spreaders, but it is generally accepted that super-spreaders are those epidemiological units (e.g. individuals, vectors, households) that account for a high proportion of the transmission, sometimes based on the 80–20 "Pareto" rule [6]. In this study a super-spreader is a house infecting two or more other houses. This means that super-spreading, as defined in our study is dependent of location and house characteristics (and the characteristics

of people living in the house). It is dependent on location because the spreading is influenced by the environment where mosquito moves. When an area contains more than one super-spreader for a certain length of time then the area is defined as hotspot.

In this work we aim to identify stable and unstable hotspots and malaria super-spreader houses by explicitly modelling infected mosquito movements from house to house. In densely populated areas the distance adult anophelines travel in search of a blood-meal are expected to be relatively short, but in sparsely populated settings, mosquitoes are likely to disperse up to several kilometres, depending upon host availability and landscape structure [8]. Household location, occupancy, demography and characteristics (e.g. building materials and entry points) influence indoor vector densities, and consequently mosquito directional movement and dispersal, the distribution of human-biting risk and malaria transmission within communities [9,10]. These factors can impact the effectiveness of interventions over a range of geographical scales.

Mosquitoes, as with many other insect vectors, use wind-assisted movement to disperse or migrate with major implications for pathogen spread and gene flow [11,12]. Over relatively short distances (hundreds of metres) mosquitoes use the wind to seek and guide themselves towards odour plumes emitted from hosts although there is little consensus under which prevalent flight strategy (e.g. upwind, downwind, random) and optimal wind speed they make such flights [13–15].

Modelling disease transmission using vector flight behaviour and wind data are a powerful means to quantify and characterise vector dispersal, determine environmental risk factors for transmission, and predict the development of an epidemic, leading to the possibility of short, medium and large spatial scale estimations of disease spread [16,17]. There are only a limited number of models explicitly accounting for flight behaviour and flight parameters when predicting *Anopheles* dispersion [18,19], and none joining *Anopheles* dynamics or mosquito wind-aided dispersion and human malaria transmission (for a review see [20]).

In this work, we have extended the biologically-informed Spatial-temporal Wind-Outbreak Trajectory Simulation model (SWOTS) to the malaria disease system (MALSWOTS). SWOTS was originally developed for describing the wind spread of midge-borne diseases (*Culicoides sp.*) in livestock [21,22] based on midges' flight capabilities and wind conditions. SWOTS estimated intrinsic and extrinsic incubation periods that corresponded with those found by laboratory analyses [22], confirming the validity of SWOTS results. Compared to the original SWOTS model, in MALSWOTS the mosquito movements between locations: (i) are weighted on the survival probabilities of mosquitoes based on the environmental characteristics of departing and arriving households and (ii) inform the inferential process by accounting for uninfected households. This provides estimations of the probability that a connection between two locations takes place by integrating likelihood-based approaches with mechanistic simulations allowing for predictions and uncertainty in those predictions. SWOTS and MALSWOTS are different from other wind-based models such as HYDREMATS [23]. The latter combines life-stage population dynamics with environmental conditions in a fully mechanistic process, while MALSWOTS allows for the probabilistic treatment of source (infected mosquito departing) and sink (infected mosquito reaching an uninfected house), and inclusion of downwind and upwind mosquito movement capabilities with or without a random component.

We apply MALSWOTS to identify super-spreader houses and clusters of these super-spreaders as hotspots in the villages surrounding the Majete Wildlife Reserve, a malaria endemic area in the Chikwawa District of southern Malawi, part of the Majete Malaria Project (MMP) [24,25]. Malaria endemicity in the study area is high but heterogeneous with identifiable hotspots of transmission [26]. Transmission of the malaria parasite *Plasmodium falciparum* occurs year-round, with annual peaks typically following the rainy season [27]. From

2015 to 2018 the prevalence of *P. falciparum* infection in children aged 6–59 mo. during repeated cross-sectional surveys varied within and between years, from a low of 5% in December 2016 to a high of 60% in June 2015 [28]. The Chikwawa District Health Office implements malaria interventions in the district for the National Malaria Control Programme (NMCP). This includes provision of ITNs to pregnant women and children under 5 years old, mass distribution campaigns of ITNs, intermittent preventative therapy for pregnant women, and malaria case diagnosis and treatment with artemisinin-based combination therapy. The only mass distribution of ITNs during the study period was in April 2016. PermaNet 2.0 (Vestergaard Frandsen, Lausanne, Switzerland), Olyset Net (Sumitomo Chemical Company, Tokyo, Japan), and Royal Sentry (Disease Control Technologies, USA) were distributed in the district.

We provide an overview of the MALSWOTS algorithm, its application to the MMP data, present various outputs and indexes returned by MALSWOTS such as identification of superspreaders, stable and unstable hotspots, velocity of parasite transmission, spatial and temporal scales of parasite transmission. In addition, we validate the model with an out-of-sample procedure and describe how MALSWOTS outputs can support and contribute to spatially targeted malaria intervention strategies.

## Methods

### Ethics statement

Written informed consent was sought from individual participants (and their parents or guardians when appropriate) during household surveys. The College of Medicine Research and Ethics Committee in Malawi approved this study (Proposal Number P.05/15/1731).

### Study area and period

Data were collected as part of the MMP in Chikwawa district, southern Malawi. The MMP assessed multidisciplinary and community-based approaches to malaria control from 2014 to 2019 in communities living within 10 km of Majete Wildlife Reserve [25,26]. MMP studies primarily covered three separate regions, referred to as focal areas A, B and C, with a total of 25,000 inhabitants living in 6,600 houses across 65 villages [26] (Fig 1). The MMP included a cluster-randomized controlled trial from May 2016 to May 2018. Of the 65 villages, 12 were excluded from the treatment arm allocation to reduce the risk of contamination between different treatment arms. The remaining villages were assigned to one of four groups: a control arm; house improvement (HI); larval source management (LSM); and HI+LSM. NMCP interventions and community engagement were used in all arms.

Chikwawa sits within the lower Shire Valley which, compared with the rest of the country, has its own unique climate characterised by hot/dry periods (September to December) and comparatively low, sporadic rainfall. The higher temperatures and presence of the Shire river promotes *Anopheles* mosquito proliferation. The dominant vectors in the region are *Anopheles funestus* and *An. arabiensis* with a small proportion of *An. gambiae* s.s. [24]. Mosquito population size is dependent on seasonal rainfall [29].

Most of the households practice subsistence farming with maize, millet and beans as staple crops [30]. Many residents also keep domestic animals, with cattle marketing forming an important economic activity in the district.

To measure malaria parasite prevalence, transmission and vector densities, a series of repeated cross-sectional surveys were conducted from 2015 to 2018. The period covered by the current analysis is 1st July 2016 to 12th May 2018 (680 days), coinciding with the collection of wind data as described below (Table 1).

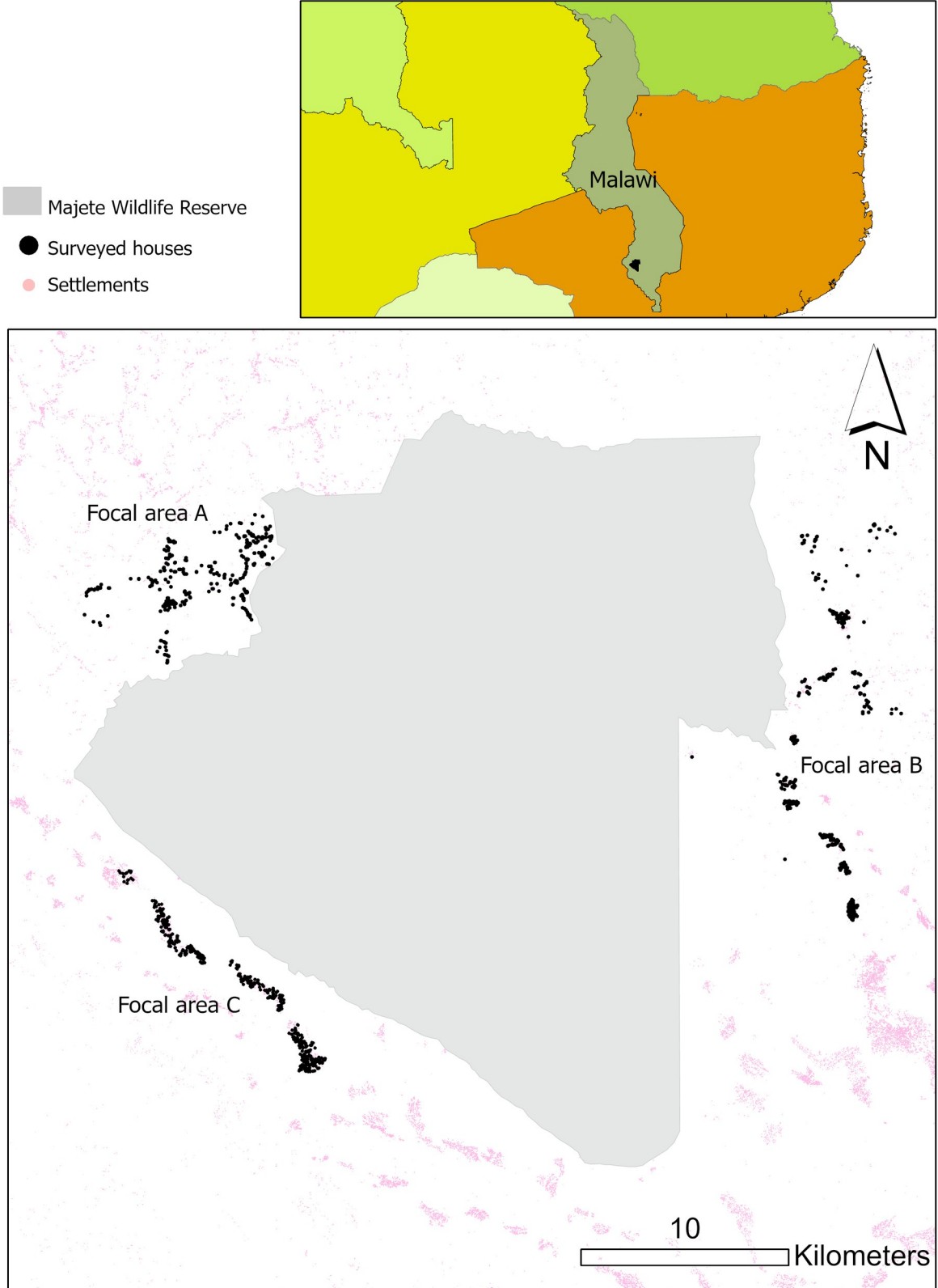

**Fig 1. Majete Malaria Project focal areas.** Map was made using ArcGIS Pro 2.7.0 (https://www.esri.com/en-us/arcgis/products/arcgis-pro/resources). Source settlements data: High Resolution Settlement Layer—Facebook Connectivity Lab and Center for International Earth Science Information Network—CIESIN—Columbia University.

**Table 1. Environmental and epidemiological characteristics of focal areas for the period under study (1st July 2016 to 12th May 2018).**

| Parameters | Focal area A | Focal area B | Focal area C |
|---|---|---|---|
| Area (Km$^2$) | 85.94 | 209.69 | 128.18 |
| Average temperature (˚C)* | 24.39 (15.39, 35.13)^ | 25.95 (14.76, 39.06)^ | 27.19 (15.70,38.73)^ |
| Average relative humidity (%)* | 68.83 (25.70, 96.90)^ | 71.53 (24.9, 98.5)^ | 64.46 (22.70, 96.80)^ |
| Average rainfall (mm)* | 0.09 (0.00, 0.40)^ | 0.07 (0.00, 0.20)^ | 0.07 (0.00, 0.40)^ |
| Average wind speed (m/s)* | 0.22 (0.00, 2.01)^ | 0.04 (0.00, 0.50)^ | 0.04 (0.00, 0.50)^ |
| Average wind direction (degrees)* | 133.04 (29.5, 300.40)^ | 150.18 (9.80, 348.20)^ | 133.60 (29.50, 280.80)^ |
| Average gusts speed (m/s)* | 1.41 (0.00, 7.55)^ | 1.16 (0.00, 4.03)^ | 1.24 (0.00, 4.03)^ |
| Total people surveyed | 1230 | 1078 | 963 |
| Number of houses | 348 | 454 | 476 |
| Number of people/house | 3.59 | 2.51 | 1.95 |
| Malaria RDT positive per people surveyed (%) | 9.59 | 8.90 | 29.91 |
| Number of uninfected houses | 254 (73%) | 358 (78%) | 290 (61%) |
| Number of houses infected only once | 86 (25%) | 65 (14%) | 181 (38%) |
| Number of reoccurring infected houses | 8 (2%) | 31 (8%) | 5 (1%) |
| Number of reoccurring infected houses within 30–60 days a.p.i. | 1 (55 days a.p.i.) | 1 (41 days a.p.i) | 1 (43 days a.p.i) |
| Median distance between infected and uninfected houses (m) | 3165 (357, 7484)^ | 6330 (206, 16289)^ | 3945 (219, 10560)^ |
| Median distance between infected houses (m) | 2976 (94, 7294)^ | 6373 (0, 16144)^ | 3889 (208, 10318)^ |
| Proportion of infected houses having the closest houses uninfected (%) | 43 | 39 | 52 |
| Geometry by L-test infected houses | Low clustering | Medium clustering | High clustering |
| Geometry by L-test infected and uninfected houses | Low clustering | Medium clustering | High clustering |

*Average of hourly values

^ 95% confidence interval; a.p.i. after previous infection.

## Climate and wind data

Climate data were obtained from a HOBO weather station (Onset Computer Corporation, Massachusetts, USA) at each focal area with geographic coordinates (latitude and longitude in WGS84 coordinate system): -15.849767 and 34.505821 (Focal area A), -15.90287 and 34.74525 (Focal area B); and -15.97293 and 34.465394 (Focal area C). The weather stations recorded total hourly rainfall in mm; hourly average temperature in degrees Celsius (˚C); hourly average relative humidity as a percentage; hourly average soil water content in m$^3$/m$^3$; hourly average wind speed in m/s; maximum 3-second gust speed at hourly intervals in m/s; and hourly average wind direction in degrees (Table 1).

## Epidemiological data

Epidemiological data were collected using repeated cross-sectional surveys adapted from the malaria indicator survey (MIS) [26]. Every two months a sample of 270 households (90 households per focal area) was selected as described previously [24] from a demographic database covering the study area, and surveys were conducted in these households over a 6- to 8-week period in each round with short breaks between rounds (5 to 15 days). Surveys were conducted in parallel in each focal area, and 11 rounds were performed between 1st July 2016 to 12th May 2018. During each visit to a household, all children aged 6–59 months and women aged 15–49 years present during the survey were tested for the presence of *P. falciparum* antigens using a Rapid Diagnostic Test (RDT) (SD BIOLINE Malaria Ag P.f. HRP-II, Standard Diagnostics, Yongin-si, Republic of Korea). In this study, a household was defined as "a social group made

up of people eating from the same pot", whereby the members of a household could occupy one or multiple houses (i.e. buildings with distinct structural features and geographic locations). For households with multiple houses, eligible children and women from all houses were invited to participate in the study. The geographic unit of infection is therefore the house and not the household, which enables us to capture fine scale vector-biological, geographical and ecological conditions related to mosquito movement. For houses presenting more than one infection during the surveyed period, we assumed that once the first infection was recorded, any subsequent infections occurring one or more months later were independent from the initial one (i.e. a new infectious mosquito had reached the house). This is also coherent with the fact that participants presenting a positive RDT were treated according to Malawi national treatment guidelines.

During the study period, 502 *P. falciparum* infections were confirmed out of 3,271 tests. The number of malaria infections, infected houses (with at least one malaria infection), uninfected houses and houses with reoccurring infections in each focal area are presented in Table 1. Note that we are defining infected or uninfected houses based only on children from 6 to 59 months and women from 15 to 49 years old. In this study, a large proportion of RDT positive participants were asymptomatic at the time of the RDT (and did not report symptoms in the last 48 hours). Since asymptomatically infected individuals do not actively seek antimalarial treatment, their infections may last longer than symptomatic episodes and this may contribute to the further spread of malaria (Drakeley, Gonçalves et al. 2018).

### Entomological data

For each round of data collection (i.e. every two months), 72% of the households selected for epidemiological data collection (i.e. 65 of 90 households in each focal area) were randomly selected for mosquito sampling. Maximum operationally feasible sample sizes for mosquito sampling (n = 65 households per focal area) and epidemiological data collection (n = 90 households per focal area) were determined during baseline rounds in 2015. A total of 4302 adult mosquito collections were carried out (1520 in Focal area A, 1407 in Focal area B and 1375 in Focal area C) averaging 65 trapping nights per month per focal area. Adult mosquitoes were sampled using the Suna trap (Biogents AG, Regensburg, Germany), which uses a chemical bait and carbon dioxide to attract mosquitoes at night [31]. Previous work in this study area has shown that the Suna trap can be considered a good proxy for host seeking mosquitoes [32]. At each house, the trap was set one night indoors and one night outdoors consecutively, with the order of indoor/outdoor determined randomly. Mosquitoes were collected from the traps each morning. All collected mosquitoes were preserved using a desiccant and identified to species using standard morphological and molecular techniques [33–35]. qPCR was used to assess the presence of *P. falciparum* sporozoites in the heads and thoraces of female *Anopheles* mosquitoes [36]. Full description of the surveillance methods are described elsewhere [26].

Overall, the number of *Anopheles* malaria vectors collected was low (S1 Table). Only one *Anopheles*, which was not positive for *P. falciparum*, was found in Focal area A. On average, in Focal area B and C one female *Anopheles* was collected every 10 and 14 trapping nights, respectively, with one *P. falciparum* positive *Anopheles* collected per 200 trapping nights in Focal area C and per 350 trapping nights in Focal area B. The survival probabilities were based on the number of female *Anopheles* per trap per day.

### Mosquito flight parameters

To control for mosquito flight characteristics and to quantify the influence of wind and mosquito flight on malaria transmission in the MMP focal areas, mosquito flight parameters were

obtained from the literature. The parameters considered in this work are provided in the form of ranges (r), thresholds (t) and fixed values (f). They are: number of days prior to a new human infection (DPI), that has been set between day 21 and day 1 before the date of the house survey (r); number of days of flight (DoF), 1 to 18 days (r); numbers of hours of continuous flight, one hour (f); maximum distance of flight in one day, 1 km (t); distance around the house, 15 m (t); upwind and downwind flight speed, 0.1 m/s (f); distance at which the mosquito can detect the host, 50 m (t); wind limits for random, upwind and downwind flight, 11 m/s (t); and tolerance around wind direction, 45 degrees (t). DPI accounts for the full intrinsic incubation period, uncompleted extrinsic incubation period (if not completed during the days of flight) and uncertainty in RDT test (S1 Fig). Full details on the chosen values are provided in the S1 File.

MALSWOTS considers three mosquito flight movements and their combinations: downwind when the mosquito flight direction is coincident to that of the carrying wind (passive flight); upwind when the mosquito flight against the wind flow and usually in response to host odours (active flight); and random, i.e. irrespective of the current wind speed and direction, or the presence of wind turbulence (active flight).

## MALSWOTS

We define a wind trajectory as the imaginary polyline a particle will follow when carried by the winds. When a trajectory describes the geographic path of an infected mosquito from an infected house to an uninfected house, and subsequently causing infection, the trajectory is called 'connection'. We have simplified the jargon, defining an 'infected house' as a house with at least one child (6 to 59 months old) or a woman (15 to 49 years old) detected positive with a RDT test and, vice versa, an 'uninfected house' with a negative RDT test.

The Spatio-temporal Wind-Outbreak Trajectories Simulation (SWOTS) described in [21] has been adapted to malaria mosquitoes and developed further to include constraints such as mosquito survival probabilities and uninfected locations. As in SWOTS, MALSWOTS assumes that each infected house contains mosquitoes that are (or will develop to) infected while moving to other houses. The number of mosquitoes moving from house to house is independent from the mosquito catches (otherwise it is not possible to model areas without or scarce entomological information, such as Focal area A). MALSWOTS assumes that at least one mosquito departs from each infected house each day. Therefore, the entomological data is used only for calculating the survival probabilities by time (not space since not enough data has been collected) for each area under study (depending on local environmental characteristics). Mosquito survival probabilities are obtained from a Poisson log-linear model based on exposure time and climate. The survival probability is then measured as expected survival rates per exposure time and climate (full model specification described in S1 File).

In MALSWOTS we abandoned the 'best fit' approach (for which exact parameters are obtained) and added an averaging component that allows us to consider house infections as results of different connections at different spatial and temporal scales. For this study we averaged the top 5% of the models ranked by largest correlation. As in SWOTS, MALSWOTS includes five types of mosquitoes' movement (downwind, upwind, random, downwind and random, and upwind and random). These are then used in conjunction with probabilities of survival to obtain the probability of (i) a house being infected from all surrounding infectious houses (since one house is likely to be infected by mosquitoes arriving from multiple houses with different movement types), (ii) a house being infected by a specific house, and (iii) a house being infected by a type of mosquito movement (S2 Fig).

In brief, MALSWOTS algorithm has four steps. Firstly, identification of parameter values (such as for DPI, DoF and time of the day considered) that maximise the correlations between

mosquito trajectories and house infections trajectories at each house for each plausible temporal scale. Once these parameters are found, the top 5% of the models, ranked by the largest correlation, are averaged (second step). In the third step, MALSWOTS simulates the outbreak by re-calculating the mosquito trajectories according to the averaged parameters and a stochastic wind field. Each infected house in each interval is associated with many different possible infected mosquito trajectories going from it. Some of these trajectories will end up at uninfected houses and eventually infect them. In the last step, MALSWOTS examines the associations between suspected movement of infected mosquitoes from infected houses to susceptible houses and identifies the most likely (highest probability) source and route of infection to each susceptible house, providing the probabilities described above.

The algorithm is written in R-language [37]. The full detailed algorithmic steps are described in S1 File. All calculations are performed in a grid for each focal area, with all grids having a resolution of 1/3 of a kilometre per cell. Specifically, in a grid of 31 by 25 nodes (775 grid nodes, 86km$^2$) for Focal area A; 31 by 61 (1891 grid nodes, 210km$^2$) for Focal area B; and 34 by 34 (1156 grid nodes, 128km$^2$) for Focal area C. Temperature, precipitation, wind speeds and directions are assumed identical at all the grid nodes in each area, since only one weather station is available per focal area. Triangulation between the three weather stations to estimate wind speed and direction at all the nodes in a grid is not feasible due to the large distance between focal areas (from 18km between Focal area A and C, 28km between Focal area B and C, and 29km between Focal area A and B).

Limitations described in the epidemiological data section are accounted for by considering temporal periods (number of days prior to a new human infection + number of days of flight) instead of exact dates. In general, issues related to temporal uncertainty are partly alleviated by the fact that MALSWOTS explores different time and spatial scales independently from each other, which can potentially balance the time-dependent and spatial-dependent RDT uncertainties within the individual-based level in a repeated randomised cross-sectional study [38].

## Houses spatial randomness

Environmentally constrained infectious disease spread is also dependent on the spatial configuration of the area and its units (e.g. houses) [9]. To estimate potential clustering in house locations (independently from their infection status), complete spatial randomness was tested by employing a K-function test which is independent from the shape of the study region and takes into account the density of the events. The latter allows for comparison among groups regardless the event prevalence [39]. In this study the K-function is transformed into an L-function to simplify interpretability. In fact, for the L-function a value of 0 means the process is completely random, while increasing positive values describe increasing spatial clustering [40]. The L-function significance is evaluated by computing its envelopes based on 999 point pattern simulations.

## Hotspot and super-spreader definitions

Our definition of a super-spreader is 'a house that is the source of infection for two or more other houses within the study period (07/2016 to 05/2018). Hotspots are usually defined statistically [28,41] although no strict definition exists. Here we simply considered a hotspot an arbitrary area of 1 km$^2$ (as the geographic size for hotspots described in [42]) containing at least two super-spreaders. We defined 'stable hotspot' if it occurred at least once a year in the same 1 km$^2$ over the three calendar years, otherwise it is defined as an 'unstable hotspot'. Low transmission areas are those that contain unclustered spreaders and super-spreaders.

## Investigating risk factors of super-spreader houses

The number of 'infected houses' originating from each house was modelled using a Poisson generalised linear mixed model. This model was not corrected for spatial autocorrelation because the outcome number of infected houses originating from each house was affected by a strong nugget effect (weak functional structure in the semivariance, see S3 Fig) [43]. For each house the following predictors and/or strata were considered: the total number of people in the household; the presence of any ITNs in the household; the mean wealth score for the household based on assets owned; the number of ITNs per person; treatment arm in the cluster-randomized controlled trial; the number of women, children under 15 years old, and children under 5 years old in the household; the number of rooms used for sleeping; the number of cattle, goats, sheep, chickens, and pigs owned by the household; the number of ITNs in the household; the material of the roof, wall, and floor; whether the eaves were open or not; whether the household owned any cattle, goats, sheep, chickens, or pigs; the number of people who slept under an ITN the previous night; whether the individual tested by RDT slept under an ITN the previous night; and the proportion of people sleeping under an ITN the previous night; and the trial treatment arm, as described in [24]. For villages with the same interventions as the control but which were excluded from the trial analysis because of proximity to other villages, treatment was defined as "excluded". Finally, the unconstrained population counts (because of the low accuracy of building satellite information for the Majete rural area) for the year 2016 at a resolution of 3arc (approximately 100m at the equator) from WorldPop [44], has been included as covariate.

The best fitting model was obtained from a model selection analysis by testing all the possible combinations of predictors with different stratification. The AICc criteria is recommended for small sample sizes and when the number of parameters among the compared models are different. The model with the lowest AICc value is usually considered to be the best data fitting model [45]. Due to computational issues and the small sample size (for example the number of infected houses in Focal area A, n = 86), we excluded from the model selection the interaction terms between predictors.

Variable selection was performed in R-cran software (4.0.4) with packages MuMIn (model selection and averaging) and stepcAIC.

## MALSWOTS validation

The algorithm was evaluated by cross-validation on 10% of houses for each focal area. In particular, we ran MALSWOTS for 90% of houses and calculated the probabilities for the remaining 10% to be infected (equivalent to the probability (i) described in the MALSWOTS section above) during the estimated DPI+DoF period prior to the RDT positive test, the period prior to DPI+DoF and the period after DPI+DoF (S1 Fig). Probability calculations are described in the section above. We define a 'correct prediction' if the probability of a house which gets infected during the DPI+DoF period is larger than the probability before or after the DPI +DoF period. The before period is defined as between $2^*$(DPI+DoF) and (DPI+DoF) days prior to a RDT positive test and the after period is defined as between (DPI+DoF) and $2^*$(DPI +DoF) days after a RDT positive test. For the uninfected houses, since these are always uninfected during the study period, we have not calculated probabilities for the period before, around or after a RDT positive test. Instead, we have calculated the probability to be infected during the whole period spanning from $2^*$(DPI+DoF) days before and after a RDT positive test related to the closest infected house. From this probability we have produced the number of false positive when considering probabilities of connections above 0.025 and 0.25.

## Results

In the surveyed area (Fig 1), 26% of surveyed houses were infected at some point during the two-year study period, and only 3.6% had recurring cases of malaria. For the latter only 1 house in each area experienced recurring cases within 30 to 60 days after the previous infection, while the rest of the recurring cases were after 121 days from previous infection (Table 1).

The MALSWOTS model found that in all focal areas, winds associated with the 6pm to 10pm time window are highly associated with the geographic spread of malaria parasites between houses surrounding the Majete Wildlife Reserve. In fact, winds within this timeframe accounted for between 39% to 57% of the top 5% of the strongest correlations between mosquito movements and malaria spread from house to house. It is worth noting that we explored 9,000 scenarios, each one based on a different combination of temporal and flight movement assumptions. For the rest of the time periods considered in this analysis, the second most important timeframe was 11pm to 3am in Focal areas A and C and 4am to 8am in Focal area B. Accounting for the time period as a whole (6pm-8am) did not generate the majority of highest correlations (Table 2). The modal number of days for a new infection to arise (that combines DPI and DoF) was in the range of 17 to 29 days (during which the mosquito took an infected blood meal, the mosquito flew to a different house, the extrinsic incubation period was completed, the mosquito took a blood meal from a new person, and the intrinsic incubation period was completed).

Wind-aided mosquito movements explained 91%, 85% and 71% of the possible house-to-house malaria transmissions in Focal area A, B and C respectively, with a probability above 0 (for example if this probability was 0.01, out of 100 infected mosquito wind-based trajectories departing from an infected house, at least 1 would reach an uninfected house and would subsequently become infected). Considering a more conservative probability of 0.25 (i.e. out of 100 infected mosquito wind-based trajectories departing from an infected house, at least 25 reach the uninfected house which subsequently becomes infected), these proportions reduced to 56%, 76% and 37%. A value of 25% of connections roughly equates to thousands of successful connections, and therefore the lower probabilities are still epidemiologically important.

In all focal areas, a mixture of wind-aided flight (downwind and downwind + random movements, upwind and upwind + random movements) were implicated in the majority of house-to-house transmissions: 96% in Focal area A, 58% in Focal area B, and 86% in Focal area C. Purely random movement (independent from wind) accounted for the majority of the connections in Focal area B only (around 42% of connections, Table 2). Importantly, active flight (upwind and upwind + random) is responsible for more than a third of connections in Focal area A and B and the majority of connections in Focal area C, supporting the epidemiological importance of upwind flight for mosquito dispersal and subsequent parasite transmission [46,47].

More than a third of infected houses infected at least one other house and more than a third infected two or more houses (Fig 2); the latter we define as 'super-spreaders'. The maximum observed number of infected houses resulting from a single house was 13 in Focal area C. The proportion of super-spreaders is between 5 and 13% of all houses (Table 2). The locations of the super-spreaders are shown in Fig 3. Often, these super-spreaders tend to be localised, forming hotspots (Fig 3). 92% of super-spreaders in Focal area A were within a 1 km$^2$ of an additional super-spreader (a condition we used for hotspot definition) while in the other two focal areas, all super-spreader houses are within 1 km of each other (Fig 3). Additionally, more than half of all super-spreader houses (57% to 62%) were stable; occurring at least once a year in the same 1 km$^2$ (in each cluster) over the three calendar years. The low transmission areas are relatively small in Focal area C (Fig 3). On average each super-spreader infected 3 other

**Table 2. MALSWOTS estimated optimal parameters and transmission summaries obtained from the top 5% of the models ranked by largest correlation.** ^ confidence intervals. All parameters are calculated over the whole study period.

| Parameters | Focal area A | Focal area B | Focal area C |
|---|---|---|---|
| House to House infections explained (%) | 90.42 | 85.41 | 70.43 |
| Connections with Prob>0.025 (%) | 79.78 | 85.41 | 51.39 |
| Connections with Prob>0.25 (%) | 56.17 | 76.04 | 37.74 |
| Days previous of infection (DPI) | 15 (13, 20)^ | 6 (1, 19)^ | 6 (2, 16)^ |
| Days of mosquito flight (DoF) | 13 (8, 16)^ | 9 (9, 15)^ | 15 (1, 17)^ |
| Days to new infection | 29 (21, 32)^ | 17 (10, 34)^ | 21 (3, 31)^ |
| Selected parameters 6pm-10pm (%) | 51.28 | 39.47 | 57.14 |
| Selected parameters 11pm-3am (%) | 28.20 | 26.31 | 32.65 |
| Selected parameters 4am-8am (%) | 20.51 | 34.21 | 10.20 |
| Selected parameters 6pm-8am (%) | 38.09 | 37.71 | 31.94 |
| Connections downwind (%) | 12.10 | 6.54 | 15.32 |
| Connections downwind + random (%) | 46.81 | 17.71 | 11.83 |
| Connections upwind (%) | 8.12 | 8.23 | 15.32 |
| Connections upwind + random (%) | 29.78 | 26.04 | 44.09 |
| Connections random (%) | 3.19 | 41.48 | 13.44 |
| Infection speed (m/days of flight) | 470.93 (194, 3288)^ | 201.96 (0, 3360)^ | 429.01 (57, 1925)^ |
| Median connection distance (km) | 4.94 (0.25, 33.29)^ | 0.34 (0, 13.81)^ | 3.94 (0.27, 13.16)^ |
| Median Euclidean distance (km) | 1.79 (0.05, 6.45)^ | 0.08 (0.00, 8.85)^ | 0.91 (0.04, 7.53)^ |
| Median connection distance from super-spreaders to houses (km) | 4.72 (0.31, 29.70)^ | 0.06 (0.00, 2.30)^ | 0.47 (0.21, 6.59)^ |
| Median temporal distance from super-spreaders to houses (days) | 12 (1, 15) | 13 (12, 15) | 13 (4, 15) |
| 90% connections within a distance of (km) | 26 | 9 | 10 |
| 75% connections within a distance of (km) | 19 | 3 | 7 |
| 50% connections within a distance of (km) | 5 | 1 | 4 |
| 90% connections distance surplus (x straight line) | 20 | 3 | 6 |
| 75% connections distance surplus (x straight line) | 6 | 2 | 4 |
| 50% connections distance surplus (x straight line) | 4 | 1 | 2 |
| Proportion of infecting houses (from infectious houses) | 43.62 | 46.87 | 34.41 |
| Proportion of super-spreaders (infecting more than 1 other houses) out of infecting houses | 56.11 | 35.55 | 53.12 |
| Proportion of super-spreaders (infecting more than 1 other houses) out of all houses | 10.55 | 5.46 | 12.72 |
| Average number of infected houses by super-spreaders | 2.96 (2.00, 6.45)^ | 3.37 (2.00, 8.00)^ | 3.00 (2.00, 8.87)^ |
| Average number of infected houses out of all houses | 0.90 (0.00, 4.67)^ | 0.85 (0.00, 5.50)^ | 0.70 (0.00, 4.00)^ |

houses and in comparison to non-super-spreaders they tend to have connections which are longer in time but not in distance (Fig 4). The Poisson Generalised Linear Model analysis on the super-spreaders showed that houses are likely to infect additional houses if they contain a larger number of positive rapid diagnostic test (RDT) results, a larger number of children under five, do not belong to the trial ('treatment–excluded') and if goats were present (Table 3). Overall, the model explained 67% of the total variance. The primary material of the wall structure is associated with a decreased risk of infecting other houses (bricks and mud

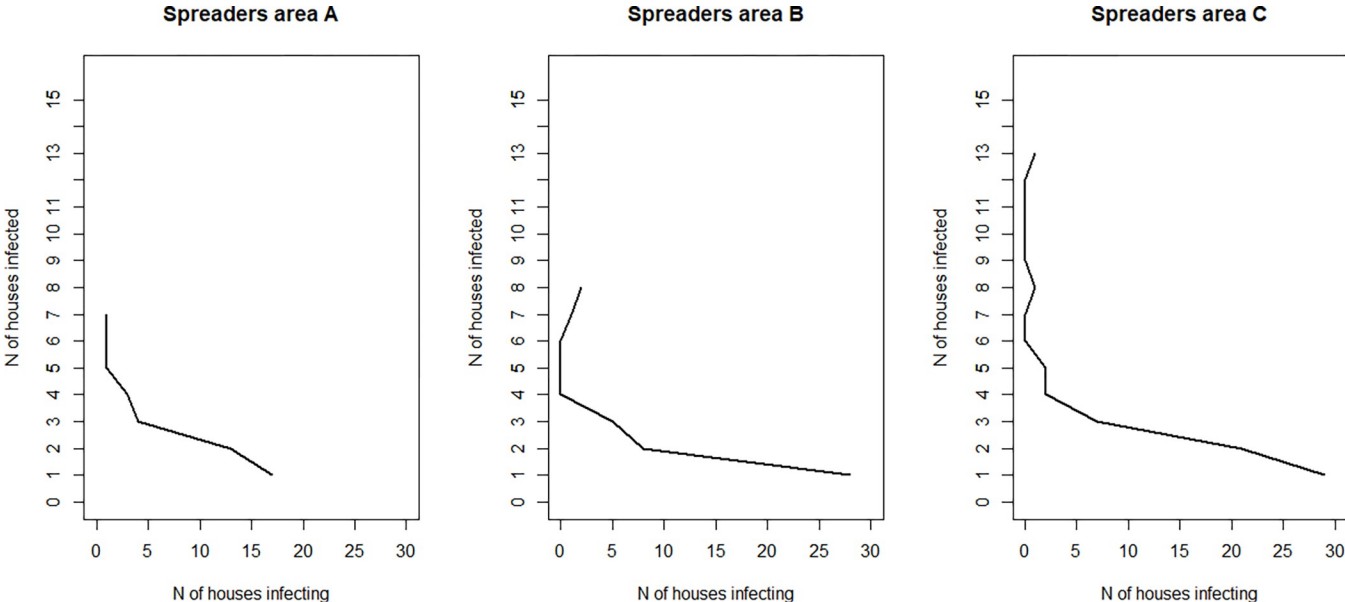

**Fig 2. Spreaders for each focal area obtained by MALSWOTS for the whole study period.** Each plot shows the number of new houses (from 1 to 15 houses in bins of 1) predicted to have been infected by *x* original houses. For example, 14 houses are likely to have infected 2 other houses each in Focal area A.

compared to other materials including iron walls), but this predictor could only be used as a random effect in the model because it was not significant as fixed effect. Population counts were not significantly associated to super-spreaders (OR 1.06, 95% confidence interval: 0.98 and 1.15), and the model explained 55% of the total variance when this predictor was included (therefore this predictor was removed from the model described in Table 3).

Successful connections are composed of relatively short daily movements that span from around 200m in Focal area B to between 400-500m in Focal area A and C. The median successful distance flown by an infected mosquito to reach an uninfected house within the optimal timespan of days prior to infection + days of flight (DPI+DoF, Table 1 for their values for each Focal area) was around 340m in Focal area B to 5 km in Focal area A (Table 2). These distances are consistently longer than the median Euclidian distance of infection connections (the straight-line distance between infecting and infected house) (Table 2). The common feature among the focal areas is that the majority of the connections are all under 5km of length (Table 2). The geometry of Focal area A differs from the other two focal areas since houses are spread much more homogeneously (see geometry parameters in Table 1 and locations in Fig 1) leading to longer connections (Fig 4). We interpret these longer distances as a result of the longer paths flown by mosquitoes to find houses given the low clustering (Table 1) under larger wind speeds recorded in the focal area (see Focal area A wind rose diagram on Fig 5, right top panel). There is an agreement between the directions of the connections (S4 Fig) and the wind directions (Fig 5 left panels) in Focal area B and C, less for Focal area A where the upwind and downwind mixed movements have a strong random component (Table 2). In Focal area A, 90% of connections are up to 20 times longer than the corresponding straight distances between the infecting and infected houses. This value is greater than that in Focal area B (90% are up to 3 times) and Focal area C (90% are up to 6 times) (Table 2).

The relative risk of becoming infected during the study period was not large, even in locations occupied by super-spreaders (Fig 6, S1 Video) since this risk is adjusted for the number of days 'spreading infected mosquitoes'. For example, a low number of super-spreaders that

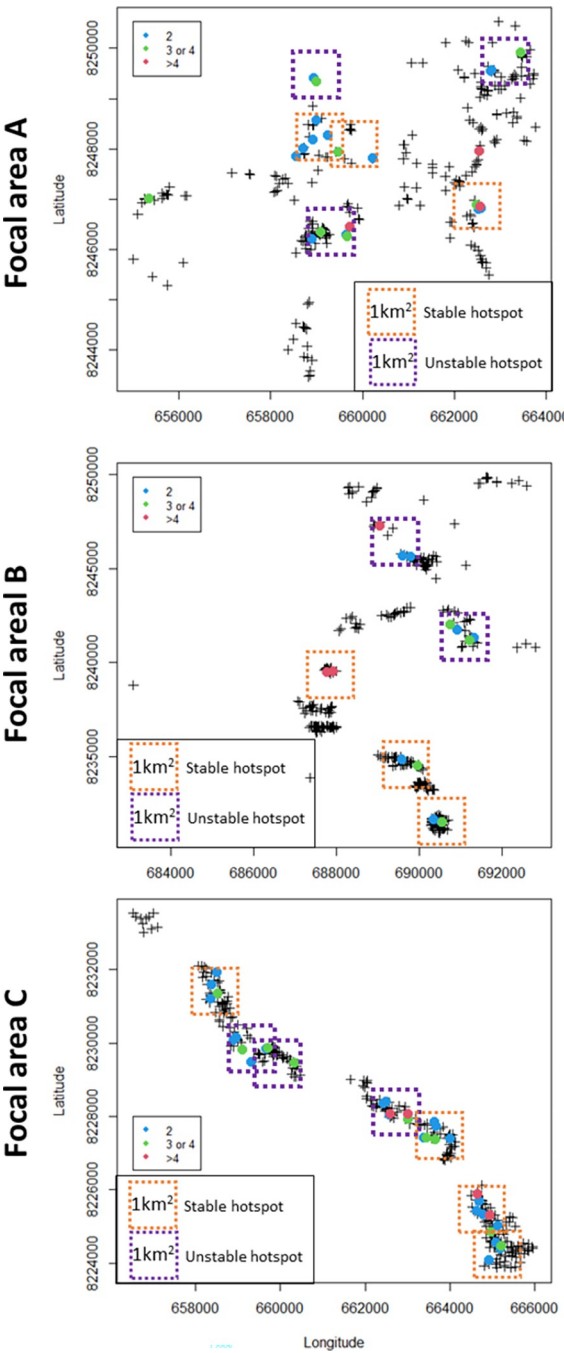

**Fig 3. Location of super-spreader houses and hotspots (stable and unstable) within each focal area.** Each colour denotes the number of 'infected houses' arising from the spreader. A black cross denotes the location of an uninfected house. Allocation of the hotspots has been performed manually capturing the largest number of super-spreaders within each hotspot square.

consistently infect other houses over time will generate a larger relative risk than a large number of super-spreaders that infect other houses for just limited periods.

Finally, the cross-validation analysis showed that MALSWOTS (Table 4) correctly predicted that the majority of those houses used for validation (10% of the original data) were infected during the DPI+DoF time before and after an RDT positive test. False positives for

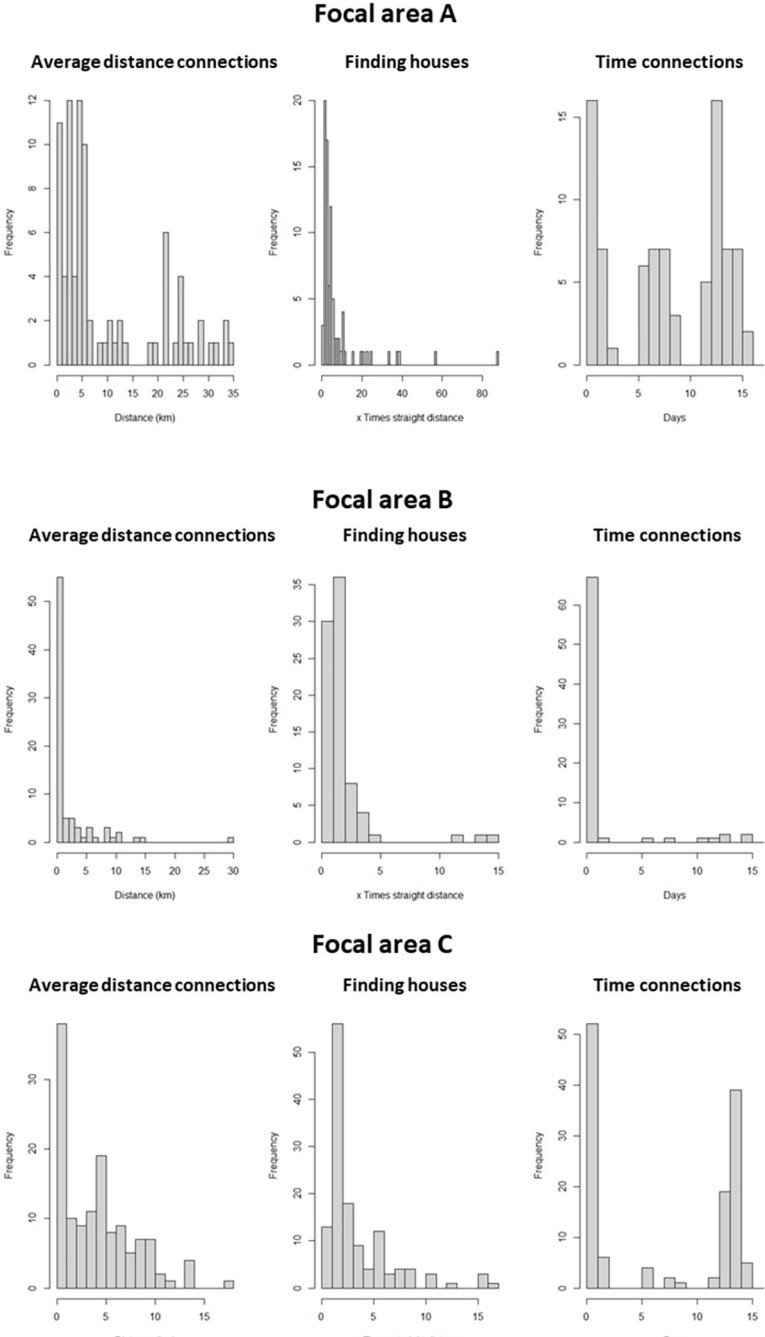

**Fig 4. MALSWOTS time and distance analyses between house connections within each focal area.** The 'average distance connections' refers to successful connections only. The 'finding houses' measure (central histogram) is the ratio between the length of the connection between two houses in which a mosquito has flown and the straight line (Euclidean) distance between the same houses (which can be shorter than the connection distance due to the zig-zag movement of the mosquito). The 'time connections' shows the temporal distance between infecting and infected houses in terms of RDT positive test. This time can be shorter than the days of flight (Table 2) due to the possibility that the mosquito left the infecting house prior malaria detection in the house.

uninfected houses were around a third in focal areas A and B but half for Focal area C. However, these values reduced to 0 when considering a more conservative probability of success of infection (Table 5). More importantly, all the super-spreaders (10% of validation houses) were

**Table 3. Odds ratios (OR) associated with selected variables from generalised linear mixed model for risk of being a super-spreader house.** *If yes, the village has the same interventions as the control village but it is excluded from the trial analysis because of proximity to other villages.

| Predictor | OR estimate | Lower 95% CI | Upper 95% CI |
|---|---|---|---|
| Number positive cases in the same house | 1.44 | 1.22 | 1.72 |
| Treatment–Excluded* (YES/NO) | 1.85 | 1.33 | 2.58 |
| Number of children under 5 | 1.22 | 0.99 | 1.51 |
| Presence of Goats (YES/NO) | 1.31 | 1.02 | 1.69 |

correctly predicted, enhancing the effectiveness of the MALSWOTS algorithm as an epidemiological forecasting tool.

## Discussion

Insect dispersal is recognised as a key determinant in the spatial distribution of vector borne disease but empirical estimates of its impact on transmission dynamics and vector control are severely lacking [5,17]. Here, we developed MALSWOTS to incorporate our current understanding of anopheline dispersal ecology to provide information on malaria transmission networks relevant to disease control.

Based on data from the MMP study, MALSWOTS identified the optimal parameter values that describe the correlation between wind data and the spread of malaria infection (from all previously infected to uninfected houses over plausible timescales) under a set of environmental conditions influencing the survival of the mosquitoes. Cross-validation demonstrated that MALSWOTS can forecast all super-spreaders and the majority of the validation houses (Table 4) once optimal parameters were identified for each area and when wind estimates for the next 16–17 days were available (Table 2). In addition, false positive uninfected houses were absent when using probability of connections above 0.25 (equivalent to 1 successful connection for every 4 potential connections).

We also explored the effect of removing survival probabilities from MALSWOTS. In terms of parameters, the major difference is the shorter DPI for Focal area A (S2 Table) than in the model with survival probabilities. Validation results show that the proportion of corrected infected houses and the probability of infection are worse in Focal area A and C but unchanged in Focal area B (S3 Table) when mosquito survival probabilities are not included. This shows that the inclusion of mosquito survival probabilities increases the accuracy of the predictions. However, this seems not the case for Focal area B, probably due to the shortest time of 'Days to new infection' among the three areas (Table 2).

One of the most important outputs from MALSWOTS is the identification of super-spreader houses from pre-existing infection data and their hotspot formation. These super-spreaders are the cause of 59% and 62% of all infections for focal areas A and C respectively, and 80% of infections for Focal area B; the latter satisfying the 'Pareto rule' [6]. The difference in super-spreader house locations across the focal areas drives a marked and seasonal spatial heterogeneity in the relative risk of malaria transmission [48] (S1, S2 and S3 Videos). Super-spreading can happen if some hosts are more infectious than others, if some hosts are more exposed to vectors (due to differences in behaviour, e.g. staying outside late, house structure, access to interventions), or because a host is preferentially bitten by more vectors [6,49]. Several biological, demographic and economic factors will determine whether members of a household are considered super-spreaders. From the MMP data available, we identified the number of positive cases in a single household, the number of children living in the house, location of the village relative to neighbouring villages (*treatment excluded)* and the presence

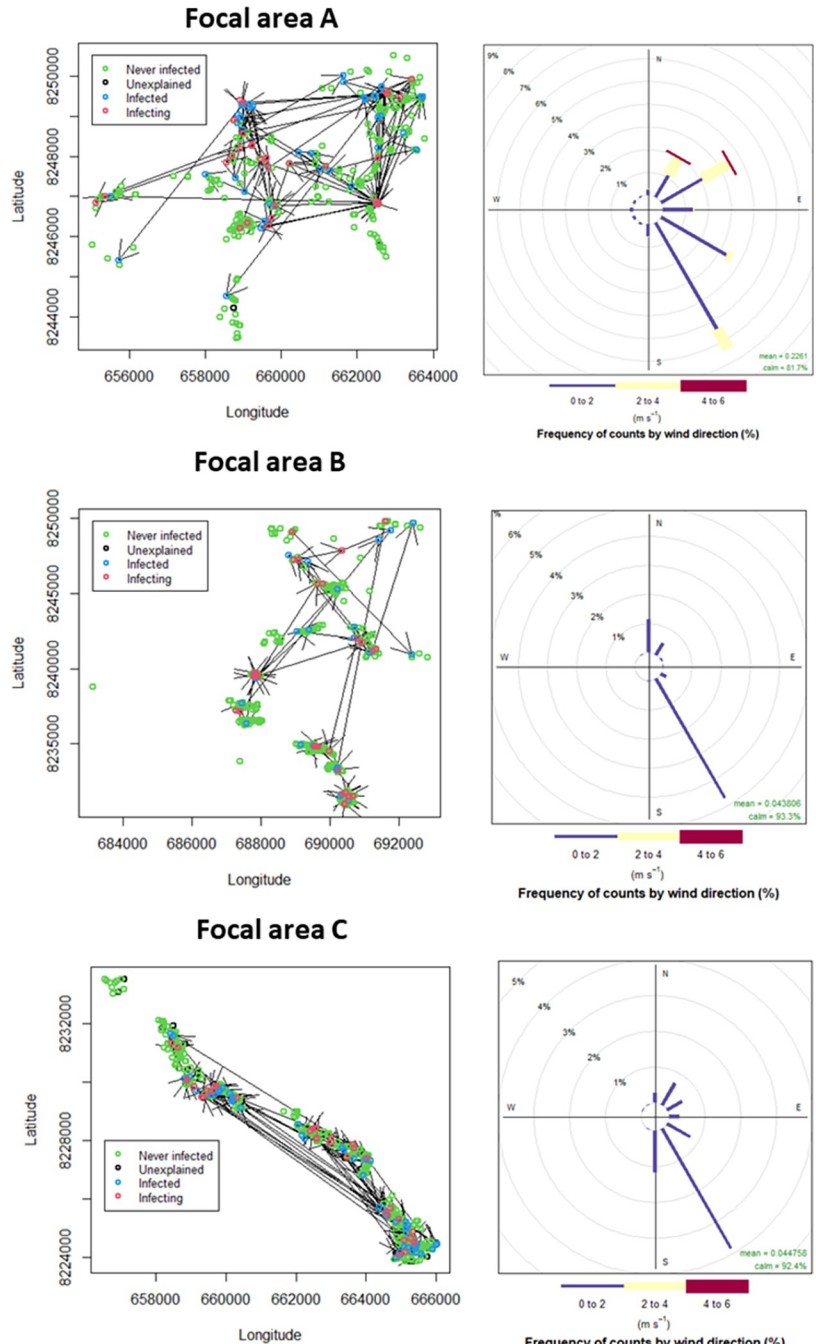

**Fig 5. Successful mosquito connections (left panels) are shown as arrows departing from an infecting house (red circle) and arriving to an uninfected (and subsequently infected) house (blue circle).** Unexplained (black circle) are houses that are not analysed because they are outside the spatial and temporal scales considered by the best models. The wind rose panels on the right show the frequency of wind directions and angular sector. The line colours identify the average speed of the winds. Calm refers to winds below 0.5 m/s.

of livestock (potentially associated to *Anopheles* abundance [50]) as significant risk factors for super-spreader houses, each of which, may have influenced the vectorial capacity of the departing and arriving infected mosquitoes [51–53]. By contrast, population counts were not significantly associated to super-spreaders, although as shown in different studies, this is linked to

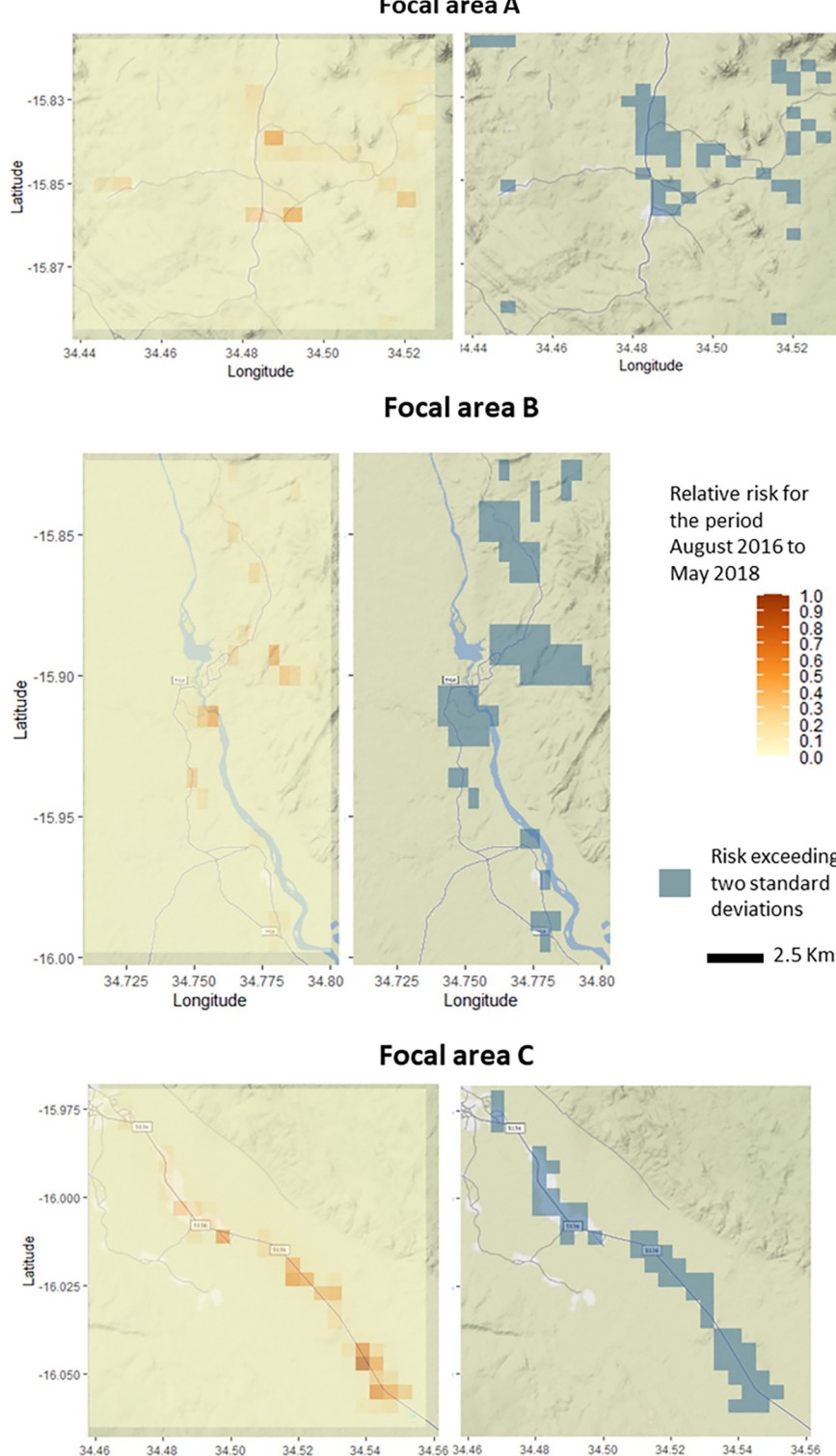

**Fig 6. Relative risk of getting infected per day (left) and risk exceedance (right) maps for each focal area.**

**Table 4. Cross validation results for infected houses.**

| Focal area | % correct prediction | Probability before | Probability around time of infection* | Probability after |
|---|---|---|---|---|
| A | 66 | 0.18 | 0.61 | 0.21 |
| B | 77 | 0.11 | 0.74 | 0.15 |
| C | 55 | 0.23 | 0.49 | 0.28 |

* The probability around time of infection is defined as the probability to be infected during one DPI+DoF before and after a RDT positive test.

malaria risk [54]. The increased risk for super-spreaders in the villages defined as 'treatment excluded' is not clear but may be due to the fact that they are centrally located within each focal area and therefore more subject to cumulative process effects (more mosquitoes reaching the centre instead of the periphery).

The importance of hotspots as intervention targets is based on three questions [42]: *'are hotspots stable over time?'*, *'do hotspots seed transmission to the rest of the focus of malaria transmission?'* and *'at what geographical resolution can hotspots be detected?'*. During our period of study (2016–2018), we found that super-spreaders tend to cluster together within 1 km$^2$ (Fig 3). This aggregation may be a conservative estimate since we found connections of up to 3 km in a day. In fact, dispersion capabilities over 1 km are the reasons why large clusters of infected houses will look like a geographic composite of hotspots (as in focal area A and C) instead of a single one.

Approximately 60% of super-spreaders were part of stable hotspots and this may support the effective implementation of more targeted control measures. In terms of elimination strategies, however, interventions must consider the full network of connections between stable, unstable hotspots and low transmission regions. In practice, however, targeting hotspots is currently difficult in most malaria transmission settings. In the few empirical studies conducted so far, there is no conclusive evidence on the real epidemiological gains obtained by targeting hotspots with, for example, either combined vector control and mass drug administration [41] or reactive case detection [5,41,52,55]. We found stable hotspots to be part of a complex network that includes unstable hotspots, and clustered and un-clustered infections (not belonging to hotspots); a system common in other hypo- and mesoendemic transmission settings [5,6]. From simulation studies, targeting malaria hotspots in systems containing connected hotspot and not-hotspot areas may not be sufficient for sustained long-term transmission reduction, and reinforcement of interventions in adjacent non-hotspots areas is potentially required [7]. Here, we describe the presence of a multitude of connected hotspots and non-hotspots in each focal area which may require larger and un-targeted interventions instead of more localised control.

The outputs from MALSWOTS provide parameters that are essential for any malaria transmission model accounting for vector movements and behaviour. *Anopheles* activity peaks at dusk and decreases around dawn/early morning in Malawi [56,57]. Historically *An. gambiae*, *An. arabiensis* and *An. funestus* were typically considered late-night biters, but evidence over

**Table 5. Cross validation results for uninfected houses.**

| Focal area | False positive (%) (infected only if connections with probability >0.025) | False positive (%) (infected only if connections with probability >0.25) | Overall probability to be infected |
|---|---|---|---|
| A | 37 | 0 | 0.044 |
| B | 32 | 0 | 0.084 |
| C | 53 | 0 | 0.038 |

the last 20 years has shown that this varies due to regional ecological differences and the influence of ITNs [58–60]. The MALSWOTS results from the MMP data show that 6pm to 10pm is the time of the day where winds are mostly correlated with temporal connection of infections, though wind dynamics at other times of the day (between 6pm and 8am) were also important. It is important to point out that 6pm to 10pm is similarly the time when communities commonly engage in outdoor evening activities and this may facilitate transmission in the proximity of the house [61,62].

The average speed of the spread of malaria parasite ranged from around 200 to 500 m/day among focal areas. This value combines prior information (e.g. flight speed and maximum distance) with wind conditions and epidemiological data. It is therefore encouraging that this spread is similar to the *Anopheles* dispersion distances reported in mark-release-recapture studies [8,63,64], and well below the limits set in the MALSWOTS algorithm for maximum distance flown in a single day if a suitable house is not found (1 km). The length of successful connections averaged a few kilometres which, again, fits with the overall dispersal capacity measured during mark-release-recapture experiments [65–67]. Downwind as well as upwind movements combined with a random component were responsible for most of the connections as shown in other studies [47]. Finally, the average time necessary to establish a successful connection between two houses (17 to 29 days) is sufficient for the completion of the extrinsic (in the mosquito) and intrinsic incubation period (in the human host) [68]. A great proportion of this time is composed by the mosquito movements.

The spatial configuration of villages surrounding Majete had a notable impact on connections between houses. Focal area C produced fewer connections despite similar output parameters to focal areas A and B. Higher clustering was observed in Focal area C promoting a larger significant component of upwind + random movement and this was effective for clustered houses over short distances (see the average distance between infected houses and the L-test for clustering in Table 1). While clustering promotes quick infections, Focal area C spatial configuration is characterised by a series of villages along the main road, requiring East-South-East winds or West-North-West winds for successful connections, and this itself may have reduced the number of connected houses. This highlights the importance of village configuration for the spread of infections [69].

The main limitation of our study is the population sampled for diagnosis with RDT, which can potentially bias the results in the case of the existence of age-related geographic patterns. In fact, the unit of infection was the house, and this was deemed 'infected' only when a child aged 6–59 months or a woman aged 15–49 years was positive for a malaria RDT. Anyone outside of these demographic categories was excluded from RDT testing. Secondly, missing infections due to untested residents of the house, low sensitivity of the RDT (limit of detection of parasite load is 50–200 parasites/μl [70]) and uncertainty around the exact moment of *P. falciparum* infection [71,72] could cause an increased spatial sparsity of infected houses (in a demographically homogenous scenario) and therefore increase the number of days and distance between connections, although cross-validation has shown accurate forecast of infected houses. Missing RDT tests for house members not included in the target population could have caused the underestimation of malaria persistence over time within a house. MALSWOTS explored persistence up to a month after previous infection at each infected house and identified best models with events happening at temporal scales lower than a month (Table 1). It is therefore unlikely that the persistence of malaria for 1 or 2 months in the same house may have played a central role in the malaria spread, while longer recurrences (after 4 months from previous infections) are likely to be independent from the previous infection.

We also recognise the limitation in the use of only one weather station for focal area and the limited number of variables used for modelling the survival of the mosquitoes. These may

have affected the capacity of the model to explain connections with very large probabilities. In addition, climate (e.g. daily temperature fluctuation) does not only influence mosquito survival but also parasite growth and development and mosquito behaviour [73,74], hence further development of MALSWOTS may include additional probabilistic components (in addition to the existing mosquito survival probability and mosquito movement probability) related to these factors. Finally, we have not investigated potential other causes of the strong spatial epidemiological and entomological heterogeneity in the three focal areas which may depend on demographics, housing characteristics, immunity, malaria interventions, mosquito behaviour (related to ecological conditions [23]) and human movements [75–77]. In particular by not considering immunological factors it is not possible to predict the stability of hotspots over time, since immunity may cause a stable hotspot to become unstable or to terminate. MALSWOTS analyses thousands of potential connections within different time and spatial scales independently from each other. This can balance time-dependent or spatially-dependent uncertainties originating from the factors not considered in this analysis and control for biases and confounding [78]. The study sites are treated as islands but migration of infected mosquitoes or people from one area to another, including areas not considered in the study, cannot be excluded. In the current model, the fully random movement component (independent from wind) may have captured some of the human movements themselves (Table 2).

MALSWOTS provided a quantitative assessment of the influence of wind on malaria transmission in a rural African setting. The model estimated the spatial-temporal pattern of malaria transmission according to the average distance and direction between infected houses as well as the percentage of *Anopheles* spp. movements due to wind-type. MALSWOTS quantified heterogeneity at a small scale including the identification of stable hotspots (at the house level) which were supported by cross-validation. At the same time, we identified a more complex network within the study area, composed of unstable hotspots and non-hotspots areas in addition to dispersal ranges varying from small to large.

MALSWOTS is a parsimonious and efficient model to forecast local transmission by mosquito movement when forecasted environmental (wind and climate) data is available. In this scenario we could simulate transmission patters assuming infections starting from a specific house, or if known which houses are infectious MALSWOTS can identify the most likely infected houses by mosquito movements. In the same way, the relative risk (as shown in Fig 6) for the entire area can be forecasted. However, the parameterisation for one area cannot be used for other areas given the intrinsic differences in urban and environmental compositions and structures in each area as shown in this study for the three focal areas analysed. Therefore, each area will require a separate parameters' inference (as described in stages 1 and 2 in the MALSWOTS algorithm flow in S1 File) before running the simulations (stage 3 and 4). To improve accuracy and predictions, MALSWOTS will be developed further in order to account for field experiment data (e.g. mark-release-recapture experiments), immunological factors, interventions, insecticide resistance and parasite genomics that will allow the identification of routes and trajectories of malaria transmission across various ecological settings.

Finally, the outputs from MALSWOTS will complement ongoing efforts to improve geospatial modelling of malaria transmission, for which information on the role of vector dispersal is largely lacking. Maintaining the effectiveness and coverage of core interventions will be key to sustaining the gains made in malaria control, but supplementary targeted vector control may become increasingly important in countries such as Malawi as malaria prevalence continues to decline and become more heterogeneous. In this scenario, accurate delineation of parasite transmission networks will become a priority for any intervention. Once parameterised, MALSWOTS can identify prevalent routes of infected mosquitoes which may impact the way we describe and delineate clusters, hotspots and not-hotspots areas. The model could also help

identify any buffer zone around an outbreak area that should be targeted to ensure most of the infective mosquitoes are removed from the environment.

## Supporting information

**S1 File. A full explanation of the mosquito flight parameters and MALSWOTS algorithm flow.**
(PDF)

**S1 Table. Entomological collection summaries for each focal area.** Data representation on the form m(q,s) where 'm' is mean number of mosquitoes collected per night in one trap x 1000, 'q' is the standard deviation x 1000, and 's' is the sum x 1000. s.l. refers to specimens that did not amplify in PCR, therefore species identification is based only on morphology. (-) and (+) indicate results of the qPCR for *P. falciparum* sporozoites.
(DOCX)

**S2 Table. MALSWOTS (without mosquito survival probabilities) estimated optimal parameters and transmission summaries obtained from the top 5% of the models ranked by largest correlation.** ^ confidence intervals. All parameters are calculated over the whole study period.
(DOCX)

**S3 Table. Cross validation results for infected houses using MALSWOTS without mosquito survival probabilities.**
(DOCX)

**S1 Fig. Timeline optimisation in MALSWOTS. During the number of days prior to infection (DPI) + days of flight (DoF) timeframe, we assume the mosquito fed on an infected individual, became infectious before arriving at the next house and then transmitted the parasite to the next house.**
(TIF)

**S2 Fig. Graphical representation of the three types of probabilities estimated by MALSWOTS.** Assuming mosquito survival probabilities and grid probability (of presence of infected houses) are the same for houses A, B and C, and given the fact that the number of simulations are the same for each house, and assuming the potential connections to be 10, the probability that house Z is infected from all surrounding infectious houses is 6/10 (1 connection from A, 3 connections from B and 2 connections from C); the probability that house Z is infected by house B is 3/10; and finally the probability that house Z is infected by downwind movement is 3/10.
(TIF)

**S3 Fig. Experimental variogram for the number of 'infected houses' originating from each house in the three areas.** This outcome has been modelled using a Poisson generalised linear mixed model. The Poisson generalised linear mixed model was not corrected for spatial autocorrelation since the experimental variograms show a large nugget effect, i.e. the semivariance does not increase with distance but tends to be stable or decrease (which means that locations further apart looks more similar that locations close to each other).
(TIF)

**S4 Fig. House to house connections by mosquito flight movements for each focal area. Each graph shows the directions of the movements, their overall proportions and the**

**proportion by covered distance.**
(TIF)

**S1 Video. Spatio-temporal relative risk videos for focal area A.** Video also available at
https://doi.org/10.17635/lancaster/researchdata/524.
(AVI)

**S2 Video. Spatio-temporal relative risk videos for focal area B.** Video also available at
https://doi.org/10.17635/lancaster/researchdata/524.
(AVI)

**S3 Video. Spatio-temporal relative risk videos for focal area C.** Video also available at
https://doi.org/10.17635/lancaster/researchdata/524.
(AVI)

## Acknowledgments

We thank Dr Monica Staniek and Dr Rod Dillon for the useful discussions at the onset of this
work.

## Author Contributions

**Conceptualization:** Luigi Sedda, Christopher M. Jones.

**Data curation:** Robert S. McCann, Alinune N. Kabaghe, Steven Gowelo, Monicah M. Mburu,
Tinashe A. Tizifa, Michael G. Chipeta, Henk van den Berg, Willem Takken, Michèle van
Vugt, Kamija S. Phiri, Julie-Anne A. Tangena.

**Formal analysis:** Luigi Sedda, Julie-Anne A. Tangena.

**Funding acquisition:** Christopher M. Jones.

**Investigation:** Julie-Anne A. Tangena.

**Methodology:** Luigi Sedda, Robert S. McCann.

**Project administration:** Christopher M. Jones.

**Software:** Russell Cain.

**Writing – original draft:** Luigi Sedda.

**Writing – review & editing:** Luigi Sedda, Robert S. McCann, Alinune N. Kabaghe, Steven
Gowelo, Monicah M. Mburu, Tinashe A. Tizifa, Michael G. Chipeta, Henk van den Berg,
Willem Takken, Michèle van Vugt, Kamija S. Phiri, Julie-Anne A. Tangena, Christopher
M. Jones.

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
