## [Decision Letter · Decision Letter 0]

14 Mar 2022

Dear Dr Jones,

Thank you very much for submitting your manuscript "Hotspots and super-spreaders: modelling fine-scale malaria parasite transmission using mosquito flight behaviour" for consideration at PLOS Pathogens. As with all papers reviewed by the journal, your manuscript was reviewed by members of the editorial board and by several independent reviewers. In light of the reviews (below this email), we would like to invite the resubmission of a significantly-revised version that takes into account the reviewers' comments.

We cannot make any decision about publication until we have seen the revised manuscript and your response to the reviewers' comments. Your revised manuscript is also likely to be sent to reviewers for further evaluation.

Sincerely,

Marcelo U. Ferreira

Guest Editor

PLOS Pathogens

Kirk Deitsch

Section Editor

PLOS Pathogens

Kasturi Haldar

Editor-in-Chief

PLOS Pathogens

orcid.org/0000-0001-5065-158X

Michael Malim

Editor-in-Chief

PLOS Pathogens

orcid.org/0000-0002-7699-2064

Reviewer's Responses to Questions

**Part I - Summary**

Reviewer #1: (No Response)

Reviewer #2: This work describes a study that characterizes the role of flying and (super) spreading in malaria infections for mosquitoes. Very few works take approaches with these kinds of parameters. Therefore, such studies have potential for contribution to the field.

In this manuscript, the data were rich since datasets were on entomological data, epidemiological and climate (wind, temperature and more), although the number of collected mosquitoes were low - more about that below.

Some aspects deserve more explanation. Most important the method for adjusting parameters and finding superspreaders and clusters, due to low numbers of collected mosquitoes and rare infection detection.

Reviewer #3: This paper presents an approach to determine the location of malaria hotspots

and super spreader houses, using statistical modelling of infectious mosquito movements from house

to house, integrating infection and wind data collected as part of a larger malaria study in the region of southern Malawi. Their model is able to determine key components of transmission of malaria in a rural setting, including stable and unstable hotspots of transmission, and identify key parameters such as the speed of spread between households, the role of wind factors, and mosquito movement parameters. It seems with this quantification control design for malaria in such a rural setting can be enhanced and optimized. I found the paper in general well-written, and clear, with interesting results presented in a coherent manner and supported by their analyses. However immunological factors which certainly play a role are not included and the differences and similarities between the different areas A, B and C could have been discussed more. Similarly the predictive ability of this modeling framework can be elaborated upon. In general I have only a few clarification points.

**Part II – Major Issues: Key Experiments Required for Acceptance**

Reviewer #1: PPATHOGENS-D-22-00125: Hotspots and super-spreaders: modelling fine-scale malaria parasite transmission using mosquito flight behaviour

In this work the authors studies hotspots of malaria transmission using entomological and epidemiological data. The authors conclude that their approach can identify prevalent routes of infected mosquitoes. The current version of the manuscript has major methodological and presentational issues that must be addressed as depicted below.

The structure of the article with the Introduction section followed by Results requires some extra explanation for clarity. The authors should briefly describe some of the concepts (connection, hotspots, infected houses…) in the Results section, otherwise it’s difficult to read in such order.

Although it is undoubtable that mosquito dispersal is an important component of malaria transmission, my main question is if hotspots and super-spreaders predictions would be different in the study area if only epidemiological data were used (i.e., with a simpler model). It seems to me that hotspots are defined around areas with infected houses and super-spreaders are those in which house members experienced more than average number of malaria episodes. Moreover, considering the low number of Anopheles mosquitoes that were captured (1 mosquito/1520 trap collections, 141/1407 and 103/1375 for Areas A, B and C, respectively—Table S1), I wonder if infections are locally or elsewhere acquired, especially in Area A where only one mosquito was collected. Please, clarify it.

The authors assume that once one infection was recorded, any subsequent infections occurring one or more months later were independent from the initial one. Hypothetically, a second infection (occurring one or more months later) could be generated by a mosquito that gets infected from either the index case or from another house member who were infected later and not surveyed. In this sense, malaria infections might be persistent over time within a house (being the sink and the source at the same time). It is not clear for me if the algorithm accounts for this hypothetical situation. Additionally, how was the mosquito survival probability estimated? Please, clarify it.

The authors say that MALSWOTS correctly predicted that the majority of those houses used for validation (10% of the original data) were infected during the DPI+DoP time. 10% of the original data also contains uninfected houses (and perhaps they are the majority of them). It seems to me that a good prediction power would be the one predicting infected as well as uninfected houses. The authors define “corrected predictions” as the probability of a house which gets infected during a specific period is larger than the probability before or after that the same period. Do the authors also evaluate the alg prediction for houses that didn’t record any infection? Please, clarify it. Also clarify Table 4 legend—"leaving out 10% of infected houses”—and define objectively what is the meaning of “probability around the time of infection”.

Table 1 presents average and standard deviations of hourly values for some measures. Note that some of them are clearly not normally distributed in such scale and so, mean and standard deviation are not very informative in these cases (e.g., mean and sd equal to 0.09 and 0.99, respectively). It would be more informative to either rescale or to present a different measure of dispersion as, for example, the 95th percentile. Also, it would help readers if the authors presented the percentage for the “number of uninfected houses”, “number of houses infected only once” and “number of reoccurring infected houses”.

According to the authors, Table 2 shows that wind within 6pm to 10pm time window accounted for between 39% to 57% of the “top 5% of the strongest correlations between mosquito movements and malaria spread from house to house” (lines 148-152). It is not clear for me if the whole Table 2 is based on the “5% of the strongest correlations”. Although this information is found on the supp material where the authors write “top correlations” and on the MALSWOTS alg description (“Take the top n”), it is not clearly specified. Please clarify it on the table legend and also on the methods section.

Also regarding Table 2, the proportion of super-spreaders—what it seems to me to be one of the main results of the paper—is informed as a proportion “out of infecting houses”. This result would be better presented as a proportion of all houses, i.e. 10-20% as informed on the text (line 180).

Figure 2 seems to not follow the most standard presentation (explanatory variable on the x-axis and response variable on the y-axis). The authors should consider inverting the axis for better interpretation of the figure.

The authors say “The locations of the super-spreaders are shown in Figure 5.” (lines 180-181). I was not able to locate super-spreaders in Figure 5—the figure legend indicates infecting houses. In the same figure, (i) it is also not stated the meaning of the “unexplained” black circles; (ii) lat and long label axis should be corrected, and; (iii) the number of plotted arrows makes it difficult to obtain any information from the figure. Also, how are downwind and upwind defined?

Regarding hotspots (Figure 3), the authors say that “92% of super-spreaders in focal area A were within a 1km radius of an additional super-spreader (a condition we used for hotspot definition).” Figure 3 presents squares and not circles. How is the position (or center) of the hotspot defined? Why do the authors affirm that low transmission areas are relatively small? With the exception of Area C, it seems to me that the majority of houses are located out of the defined hotspots.

How do the authors measure effectives for the wind-aided/random flights (line 172)?

Why could the primary material of the wall structure only be used as a random effect in the model? (lines 193-194)

I was not able to open supp videos 1, 2 and 3.

Line 212: Should it be Figure 4 instead of Figure 2?

Line 218: What do the authors mean by “distance between two houses”? Would it be the mean distance between two randomly selected houses? Please, be more specific.

Line 309: Reference out of standard.

Line 523-524: Please, indicate the motivation to use the K-funtion test.

Reviewer #2: As already described, the datasets provide rich information for understanding clusters and the role of flying in dissemination of P. falciparum. However, the low number of collected mosquitoes and limited detection of infection in field mosquitoes is usually the rule in these entomological studies. The algorithms for detection of clusters and quantifying the flight movements are interesting but seem to require a good amount of information to be able to do estimations. The results still require some convincing evidence connecting the number of collections and the inferences. My main concerns are with a need in the Results to provide this link between entomological data and the results drawn from Figure 5. Also, the methodology should provide how the algorithm accounts for this kind of entomological dataset.

The maps do not show density of human population. A simple explanation for clustering cases would be spots with human presence, in which a transmission from one house to more than one house should be expected. I recommend more investigation on the role of density of human population.

The manuscript is written with methodology in the end. This made it more difficult to understand. Efforts are necessary to make a better flow and understanding of the results.

The term "super-spreader" tends to be overused and might have different interpretations. In this manuscript, the Introduction describes the super-spreader as accounting for a high proportion of the transmission. My perception from the analysis in Table 2 is that

super-spreader means infecting more than one house. The, in the Results section, the definition becomes when infecting two or more houses. This should be stated clearly in the Introduction and be consistent throughout the manuscript.

In the Results section, what is the probability in the less conservative scenario as opposed to the probability of 0.25?

The assumed probability in the previous scenario should be stated.

In Table 3, what means "Treatment - out"?

I suggest changing "Goats present" to "Presence of Goats (YES/NO)". If it is number of goats, please define it.

Many authors use DPI as "days post-infection". In this manuscript, however, DPI means "days of previous infection". What is the definition of DPI? Days between adult emergence and the event of infection?

What is DoP? I see in the Table, DoF, Days of mosquito Flight.

Figure S1 gives a notion of these time intervals. Still, some values seem high. For instance, 29 days to new infection would mean a mean generation time of 29 days? It seems a long interval and should be discussed.

In the methodology, why these number of 72% of houses were randomly selected for mosquito sampling?

I understand that there was low number of collected mosquitoes, but survival probability were based on overall numbers of male and female, and there is a difference in the literature between survival between male and female mosquitoes. How were survival probabilities estimated? Please review the survival probabilities and how it impacts the results.

What is "set between day 21 and day 1 before the house survey"?

Also, I expect some dependence on temperature for some parameters, such as the extrinsic incubation period.

This should be discussed.

Reviewer #3: (No Response)

**Part III – Minor Issues: Editorial and Data Presentation Modifications**

Reviewer #1: (No Response)

Reviewer #2: Abstract

The abstract is purely descriptive, without providing any of the assessments, which I think could be provided.

Table 1

Character "^" signals confidence intervals but there is only one number in parentheses.

Figure 5

What is calm (bottom right)?

Reviewer #3: 1. In my view the introduction deserves a bit more background on the epidemiology of malaria in the region that is studied, in southern Malawi or Malawi in general. What are the trends, what are the control options that are being implemented, what are the numbers of malaria. This context is needed. It is partly present in the Methods, but needs to be in the Introduction as well.

2. Table 1 needs to have units for the numbers of infections reported. If the study was conduced over two years, than these numbers need to be interpreted within this time-frame. In a first reading, I found it confusing, and hard to interpret, whether this referred to a snapshot in time or an average over a longer period.

3. I am not sure the hotspot definition is very clear in the paper. Does it have to do with deviation from the average? Mathematically, what is a hotspot? Please define it clearly.

4. The definition of super-spreaders is presented in lines 177-178, an infected house causing 2 or more than 2 other infections, but what is the average? I see the distributions in Figure 2 (which are a model-derived quantity, right?) but what is the mean? Please also always put the unit of time in these numbers. What is the spatial and temporal denominator of these infections?

5. In the part about the investigation of risk factors for super-spreaders, what is the assumption about the space? Is this a non-spatial Poisson process or a spatial one?

6. Are the super-spreaders intrinsically super-spreaders or they are a consequence, a side-effect perhaps of best environmental conditions meeting there for optimal transmission? It seems the paper is suggesting the second interpretation, which in my view violates somehow the definition of a super-spreader, it is more of a “passive” super-spreading than of “active super-spreading”. I think this notion is more accurately described by the concept of hostpots - localized- than “super-spreading” ability which somehow should be independent of space. Please comment or clarify.

7. In Figure 5, can the left panels be represented in terms of their angular distribution and matched with the wind direction frequency distribution on the right for comparison?

8. Immunological aspects are not touched upon at all in the study. Are the identified superspreaders always super-spreaders over time? Don't these families acquire immunity to malaria and then stop being super-spreaders at some point? Any comments on this phenomenon, perhaps happening over longer timescales than the ones modelled in the study would be relevant, and strengthen the paper. How could immunological factors of the human population be integrated?

9. How could this model be used for epidemiological forecasting? If we know the number and distribution of infected houses, mosquito distribution, wind directions and other variables extracted as significant here, can we apply this model to predict disease risk over time across space? I found this aspect a bit under-explored in the paper, for example connecting the differences and similarities between areas A B and C. Can the parameters estimated and reported here be (universally) used for other regions and rural settings, if we know their local variables?

PLOS authors have the option to publish the peer review history of their article (what does this mean?). If published, this will include your full peer review and any attached files.

Reviewer #1: No

Reviewer #2: No

Reviewer #3: No
---

## [Editor Report · Decision Letter 1]

27 May 2022

Dear Jones,

We are pleased to inform you that your manuscript 'Hotspots and super-spreaders: modelling fine-scale malaria parasite transmission using mosquito flight behaviour' has been provisionally accepted for publication in PLOS Pathogens.

Best regards,

Marcelo U. Ferreira

Guest Editor

PLOS Pathogens

Kirk Deitsch

Section Editor

PLOS Pathogens

Kasturi Haldar

Editor-in-Chief

PLOS Pathogens

orcid.org/0000-0001-5065-158X

Michael Malim

Editor-in-Chief

PLOS Pathogens

orcid.org/0000-0002-7699-2064

The authors have presented a very detailed, point-by-point response to all concerns and suggestions. Typos and other mistakes were corrected and the overall discussion was substantially improved.
---

## [Editor Report · Acceptance letter]

23 Jun 2022

Dear Dr Jones,

We are delighted to inform you that your manuscript, "Hotspots and super-spreaders: modelling fine-scale malaria parasite transmission using mosquito flight behaviour," has been formally accepted for publication in PLOS Pathogens.

Best regards,

Kasturi Haldar

Editor-in-Chief

PLOS Pathogens

orcid.org/0000-0001-5065-158X

Michael Malim

Editor-in-Chief

PLOS Pathogens

orcid.org/0000-0002-7699-2064